# Efficiently Computing Nash Equilibria in Adversarial Team Markov Games

**Fivos Kalogiannis**
UC Irvine

**Ioannis Anagnostides**
Carnegie Mellon University

**Ioannis Panageas***
UC Irvine

**Emmanouil V. Vlatakis-Gkaragkounis**
Columbia University

**Vaggos Chatziafratis**
UC Santa Cruz

**Stelios Stavroulakis**
UC Irvine

## Abstract

Computing Nash equilibrium policies is a central problem in multi-agent reinforcement learning that has received extensive attention both in theory and in practice. However, in light of computational intractability barriers in general-sum games, provable guarantees have been thus far either limited to fully competitive or cooperative scenarios, or impose strong assumptions that are difficult to meet in most practical applications. In this work, we depart from those prior results by investigating infinite-horizon *adversarial team Markov games*, a natural and well-motivated class of games in which a team of identically-interested players—in the absence of any explicit coordination or communication—is competing against an adversarial player. This setting allows for a unifying treatment of zero-sum Markov games and Markov potential games, and serves as a step to model more realistic strategic interactions that feature both competing and cooperative interests. Our main contribution is the first algorithm for computing stationary $\epsilon$-approximate Nash equilibria in adversarial team Markov games with computational complexity that is polynomial in all the natural parameters of the game, as well as $1/\epsilon$. The proposed algorithm is based on performing independent policy gradient steps for each player in the team, in tandem with best responses from the side of the adversary; in turn, the policy for the adversary is then obtained by solving a carefully constructed linear program. Our analysis leverages non-standard techniques to establish the KKT optimality conditions for a nonlinear program with nonconvex constraints, thereby leading to a natural interpretation of the induced Lagrange multipliers.

## 1 Introduction

Multi-agent reinforcement learning (MARL) offers a principled framework for analyzing competitive interactions in dynamic and stateful environments in which agents' actions affect both the state of the world and the rewards of the other players. Strategic reasoning in such complex multi-agent settings has been guided by game-theoretic principles, leading to many recent landmark results in benchmark domains in AI (Bowling et al., 2015; Silver et al., 2017; Vinyals et al., 2019; Moravčík et al., 2017; Brown & Sandholm, 2019; 2018; Brown et al., 2020; Perolat et al., 2022). Most of these remarkable advances rely on scalable and decentralized algorithms for computing *Nash equilibria* (Nash, 1951)—a standard game-theoretic notion of rationality—in two-player zero-sum games.

Nevertheless, while single-agent RL has enjoyed rapid theoretical progress over the last few years (*e.g.*, see (Jin et al., 2018; Agarwal et al., 2020; Li et al., 2021; Luo et al., 2019; Sidford et al., 2018), and references therein), a comprehensive understanding of the multi-agent landscape still remains elusive. Indeed, provable guarantees for efficiently computing Nash equilibria have been thus far limited to either fully competitive settings, such as two-player zero-sum games (Daskalakis et al., 2020; Wei et al., 2021; Sayin et al., 2021; Cen et al., 2021; Sayin et al., 2020; Condon, 1993), or environments in which agents are striving to coordinate towards a common global objective (Claus

---

*Correspondence to `ipanagea@ics.uci.edu`.

& Boutilier, 1998; Wang & Sandholm, 2002; Leonardos et al., 2021; Ding et al., 2022; Zhang et al., 2021b; Chen et al., 2022; Maheshwari et al., 2022; Fox et al., 2022).

However, many real-world applications feature both shared and competing interests between the agents. Efficient algorithms for computing Nash equilibria in such settings are much more scarce, and typically impose restrictive assumptions that are difficult to meet in most applications (Hu & Wellman, 2003; Bowling, 2000). In fact, even in *stateless* two-player (normal-form) games, computing approximate Nash equilibria is computationally intractable (Daskalakis et al., 2009; Rubinstein, 2017; Chen et al., 2009; Etessami & Yannakakis, 2010)—subject to well-believed complexity-theoretic assumptions. As a result, it is common to investigate equilibrium concepts that are more permissive than Nash equilibria, such as *coarse correlated equilibria (CCE)* (Aumann, 1974; Moulin & Vial, 1978). Unfortunately, recent work has established strong lower bounds for computing even approximate (stationary) CCEs in turn-based stochastic two-player games (Daskalakis et al., 2022; Jin et al., 2022). Those negative results raise a central question:

> *Are there natural multi-agent environments incorporating both*
> *competing and shared interests for which we can establish*     (★)
> *efficient algorithms for computing (stationary) Nash equilibria?*

Our work makes concrete progress in this fundamental direction. Specifically, we establish the first efficient algorithm leading to Nash equilibria in *adversarial team Markov games*, a well-motivated and natural multi-agent setting in which a team of agents with a common objective is facing a competing adversary.

## 1.1 OUR RESULTS

Before we state our main result, let us first briefly introduce the setting of adversarial team Markov games; a more precise description is deferred to Section 2.1. To address Question (★), we study an infinite-horizon Markov (stochastic) game with a finite state space $\mathcal{S}$ in which a team of agents $\mathcal{N}_A \coloneqq [n]$ with a common objective function is competing against a *single* adversary with opposing interests. Every agent $k \in [n]$ has a (finite) set of available actions $\mathcal{A}_k$, while $\mathcal{B}$ represents the adversary's set of actions. We will also let $\gamma \in [0, 1)$ be the *discounting factor*. Our goal will be to compute an (approximate) Nash equilibrium; that is, a strategy profile so that no player can improve via a unilateral deviation (see Definition 2.1). In this context, our main contribution is the first polynomial time algorithm for computing Nash equilibria in adversarial team Markov games:

**Theorem 1.1** (Informal). *There is an algorithm (IPGMAX) that, for any $\epsilon > 0$, computes an $\epsilon$-approximate stationary Nash equilibrium in adversarial team Markov games, and runs in time*

$$\mathsf{poly}\left(|\mathcal{S}|, \sum_{k=1}^{n} |\mathcal{A}_k| + |\mathcal{B}|, \frac{1}{1-\gamma}, \frac{1}{\epsilon}\right).$$

A few remarks are in order. First, our guarantee significantly extends and unifies prior results that only applied to either *two-player* zero-sum Markov games or to *Markov potential games*; both of those settings can be cast as special cases of adversarial team Markov games (see Section 2.3). Further, the complexity of our algorithm, specified in Theorem 1.1, scales only with $\sum_{k \in \mathcal{N}_A} |\mathcal{A}_k|$ instead of $\prod_{k \in \mathcal{N}_A} |\mathcal{A}_k|$, bypassing what is often referred to as the *curse of multi-agents* (Jin et al., 2021). Indeed, viewing the team as a single "meta-player" would induce an action space of size $\prod_{k \in \mathcal{N}_A} |\mathcal{A}_k|$, which is *exponential* in $n$ even if each agent in the team has only two actions. In fact, our algorithm operates without requiring any (explicit) form of coordination or communication between the members of the team (beyond the structure of the game), a feature that has been motivated in practical applications (von Stengel & Koller, 1997). Namely, scenarios in which communication or coordination between the members of the team is either overly expensive, or even infeasible; for an in depth discussion regarding this point we refer to (Schulman & Vazirani, 2017).

## 1.2 OVERVIEW OF TECHNIQUES

To establish Theorem 1.1, we propose a natural and decentraliezd algorithm we refer to as *Independent Policy GradientMax* (IPGMAX). IPGMAX works in turns. First, each player in the team performs one independent policy gradient step on their value function with an appropriately selected

learning rate $\eta > 0$. In turn, the adversary best responds to the current policy of the team. This exchange is repeated for a sufficiently large number of iterations $T$. Finally, IPGMAX includes an auxiliary subroutine, namely AdvNashPolicy(), which computes the Nash policy of the adversary; this will be justified by Proposition 1.1 we describe below.

Our analysis builds on the techniques of Lin et al. (2020)—developed for the saddle-point problem $\min_{\boldsymbol{x} \in \mathcal{X}} \max_{\boldsymbol{y} \in \mathcal{Y}} f(\boldsymbol{x}, \boldsymbol{y})$—for characterizing GDMAX. Specifically, GDMAX consists of performing gradient descent steps, specifically on the function $\phi(\boldsymbol{x}) := \max_{\boldsymbol{y} \in \mathcal{Y}} f(\boldsymbol{x}, \boldsymbol{y})$. Lin et al. (2020) showed that GDMAX converges to a point $(\hat{\boldsymbol{x}}, \boldsymbol{y}^*(\hat{\boldsymbol{x}}))$ such that $\hat{\boldsymbol{x}}$ is an approximate first-order stationary point of the *Moreau envelope* (see Definition 3.1) of $\phi(\boldsymbol{x})$, while $\boldsymbol{y}^*(\hat{\boldsymbol{x}})$ is a best response to $\hat{\boldsymbol{x}}$. Now if $f(\boldsymbol{x}, \cdot)$ is *strongly-concave*, one can show (by Danskin's theorem) that $(\hat{\boldsymbol{x}}, \boldsymbol{y}^*(\boldsymbol{x}))$ is an approximate first-order stationary point of $f$. However, our setting introduces further challenges since the value function $V_{\boldsymbol{\rho}}(\boldsymbol{\pi}_{\text{team}}, \boldsymbol{\pi}_{\text{adv}})$ is nonconvex-nonconcave.

For this reason, we take a more refined approach. We first show in Proposition 3.1 that IPGMAX is guaranteed to converge to a policy profile $(\hat{\boldsymbol{\pi}}_{\text{team}}, \cdot)$ such that $\hat{\boldsymbol{\pi}}_{\text{team}}$ is an $\epsilon$-nearly stationary point of $\max_{\boldsymbol{\pi}_{\text{adv}}} V_{\rho}(\boldsymbol{\pi}_{\text{team}}, \boldsymbol{\pi}_{\text{adv}})$. Then, the next key step and the crux of the analysis is to show that $\hat{\boldsymbol{\pi}}_{\text{team}}$ *can be extended to an $O(\epsilon)$-approximate Nash equilibrium policy:*

**Proposition 1.1** (Informal). If $\hat{\boldsymbol{\pi}}_{\text{team}}$ is an $\epsilon$-nearly stationary point of $\max_{\boldsymbol{\pi}_{\text{adv}}} V_{\rho}(\boldsymbol{\pi}_{\text{team}}, \boldsymbol{\pi}_{\text{adv}})$, there exists a policy for the adversary $\hat{\boldsymbol{\pi}}_{\text{adv}}$ so that $(\hat{\boldsymbol{\pi}}_{\text{team}}, \hat{\boldsymbol{\pi}}_{\text{adv}})$ is an $O(\epsilon)$-approximate Nash equilibrium.

In the special case of normal-form games, a similar extension theorem was recently obtained by Anagnostides et al. (2023). In particular, that result was derived by employing fairly standard linear programming techniques. In contrast, our more general setting introduces several new challenges, not least due to the nonconvexity-nonconcavity of the objective function.

Indeed, our analysis leverages more refined techniques stemming from nonlinear programming. More precisely, while we make use of standard policy gradient properties, similar to the single-agent MDP setting (Agarwal et al., 2021; Xiao, 2022), our analysis does not rely on the so-called *gradient-dominance* property (Bhandari & Russo, 2019), as that property does not hold in a team-wise sense. Instead, inspired by an alternative proof of Shapley's theorem (Shapley, 1953) for two-person zero-sum Markov games (Filar & Vrieze, 2012, Chapter 3), we employ mathematical programming. One of the central challenges is that the induced nonlinear program has a set of nonconvex constraints. As such, even the existence of (nonnegative) Lagrange multipliers satisfying the KKT conditions is not guaranteed, thereby necessitating more refined analysis techniques.

To this end, we employ the *Arrow-Hurwiz-Uzawa constraint qualification* (Theorem A.1) in order to establish that the local optima are contained in the set of KKT points (Corollary B.1). Then, we leverage the structure of adversarial team Markov games to characterize the induced Lagrange multipliers, showing that a subset of these can be used to establish Proposition 1.1; incidentally, this also leads to an efficient algorithm for computing a (near-)optimal policy of the adversary. Finally, we also remark that controlling the approximation error—an inherent barrier under policy gradient methods—in Proposition 1.1 turns out to be challenging. We bypass this issue by constructing "relaxed" programs that incorporate some imprecision in the constraints. A more detailed overview of our algorithm and the analysis is given in Section 3.

## 2 PRELIMINARIES

In this section, we introduce the relevant background and our notation. Section 2.1 describes adversarial team Markov games. Section 2.2 then defines some key concepts from multi-agent MDPs, while Section 2.3 describes a generalization of adversarial team Markov games, beyond identically-interested team players, allowing for a richer structure in the utilities of the team—namely, adversarial Markov potential games.

**Notation.** We let $[n] := \{1, \ldots, n\}$. We use superscripts to denote the (discrete) time index, and subscripts to index the players. We use boldface for vectors and matrices; scalars will be denoted by lightface variables. We denote by $\|\cdot\| := \|\cdot\|_2$ the Euclidean norm. For simplicity in the exposition, we may sometimes use the $O(\cdot)$ notation to suppress dependencies that are polynomial in the natural parameters of the game; precise statements are given in the Appendix. For the convenience of the reader, a comprehensive overview of our notation is given in A.3.

## 2.1 Adversarial Team Markov Games

An *adversarial team Markov game* (or an adversarial team *stochastic* game) is the Markov game extension of static, normal-form adversarial team games (Von Stengel & Koller, 1997). The game is assumed to take place in an infinite-horizon discounted setting in which a team of identically-interested agents gain what the adversary loses. Formally, the game $\mathcal{G}$ is represented by a tuple $\mathcal{G} = (\mathcal{S}, \mathcal{N}, \mathcal{A}, \mathcal{B}, r, \mathbb{P}, \gamma, \rho)$ whose components are defined as follows.

- $\mathcal{S}$ is a finite and nonempty set of *states*, with cardinality $S \coloneqq |\mathcal{S}|$;
- $\mathcal{N}$ is the set of players, partitioned into a set of $n$ team agents $\mathcal{N}_A \coloneqq [n]$ and a single *adversary*
- $\mathcal{A}_k$ is the action space of each player in the team $k \in [n]$, so that $\mathcal{A} \coloneqq \bigtimes_{k \in [n]} \mathcal{A}_k$, while $\mathcal{B}$ is the action space of the adversary. We also let $A_k \coloneqq |\mathcal{A}_k|$ and $B \coloneqq |\mathcal{B}|$;[1]
- $r : \mathcal{S} \times \mathcal{A} \times \mathcal{B} \to (0,1)$ is the (deterministic) instantaneous *reward function*[2] representing the (normalized) payoff of the adversary, so that for any $(s, \boldsymbol{a}, b) \in \mathcal{S} \times \mathcal{A} \times \mathcal{B}$,

$$r(s, \boldsymbol{a}, b) + \sum_{k=1}^{n} r_k(s, \boldsymbol{a}, b) = 0, \tag{1}$$

  and for any $k \in [n]$,

$$r_k(s, \boldsymbol{a}, b) = r_{\text{team}}(s, \boldsymbol{a}, b). \tag{2}$$

- $\mathbb{P} : \mathcal{S} \times \mathcal{A} \times \mathcal{B} \to \Delta(\mathcal{S})$ is the *transition probability function*, so that $\mathbb{P}(s'|s, \boldsymbol{a}, b)$ denotes the probability of transitioning to state $s' \in \mathcal{S}$ when the current state is $s \in \mathcal{S}$ under the action profile $(\boldsymbol{a}, b) \in \mathcal{A} \times \mathcal{B}$;
- $\gamma \in [0,1)$ is the *discount factor*; and
- $\boldsymbol{\rho} \in \Delta(\mathcal{S})$ is the *initial state distribution* over the state space. We will assume that $\boldsymbol{\rho}$ is full-support, meaning that $\rho(s) > 0$ for all $s \in \mathcal{S}$.

In other words, an adversarial team Markov game is a subclass of general-sum infinite-horizon multi-agent discounted MDPs under the restriction that all but a single (adversarial) player have identical interests (see (2)), and the game is globally zero-sum—in the sense of (1). As we point out in Section 2.3, (2) can be relaxed in order to capture *(adversarial) Markov potential games* (Definition 2.2), without qualitatively altering our results.

## 2.2 Policies, Value Function, and Nash Equilibria

**Policies.** A *stationary*—that is, time-invariant—policy $\boldsymbol{\pi}_k$ for an agent $k$ is a function mapping a given state to a distribution over available actions, $\boldsymbol{\pi}_k : \mathcal{S} \ni s \mapsto \boldsymbol{\pi}_k(\cdot|s) \in \Delta(\mathcal{A}_k)$. We will say that $\boldsymbol{\pi}_k$ is *deterministic* if for every state there is some action that is selected with probability 1 under policy $\boldsymbol{\pi}_k$. For convenience, we will let $\Pi_{\text{team}} : \mathcal{S} \to \Delta(\mathcal{A})$ and $\Pi_{\text{adv}} : \mathcal{S} \to \Delta(\mathcal{B})$ denote the policy space for the team and the adversary respectively. We may also write $\Pi : \mathcal{S} \to \Delta(\mathcal{A}) \times \Delta(\mathcal{B})$ to denote the joint policy space of all agents.

**Direct Parametrization.** Throughout this paper we will assume that players employ *direct policy parametrization*. That is, for each player $k \in [n]$, we let $\mathcal{X}_k \coloneqq \Delta(\mathcal{A}_k)^S$ and $\boldsymbol{\pi}_k = \boldsymbol{x}_k$ so that $x_{k,s,a} = \boldsymbol{\pi}_k(a|s)$. Similarly, for the adversary, we let $\mathcal{Y} \coloneqq \Delta(\mathcal{B})^S$ and $\boldsymbol{\pi}_{\text{adv}} = \boldsymbol{y}$ so that $y_{s,a} = \boldsymbol{\pi}_{\text{adv}}(a|s)$. (Extending our results to other policy parameterizations, such as soft-max (Agarwal et al., 2021), is left for future work.)

**Value Function.** The *value function* $V_s : \Pi \ni (\boldsymbol{\pi}_1, \ldots, \boldsymbol{\pi}_n, \boldsymbol{\pi}_{\text{adv}}) \mapsto \mathbb{R}$ is defined as the expected cumulative discounted reward received by the adversary under the joint policy $(\boldsymbol{\pi}_{\text{team}}, \boldsymbol{\pi}_{\text{adv}}) \in \Pi$ and the initial state $s \in \mathcal{S}$, where $\boldsymbol{\pi}_{\text{team}} \coloneqq (\boldsymbol{\pi}_1, \ldots, \boldsymbol{\pi}_n)$. In symbols,

$$V_s(\boldsymbol{\pi}_{\text{team}}, \boldsymbol{\pi}_{\text{adv}}) \coloneqq \mathbb{E}_{(\boldsymbol{\pi}_{\text{team}}, \boldsymbol{\pi}_{\text{adv}})} \left[ \sum_{t=0}^{\infty} \gamma^t r(s^{(t)}, \boldsymbol{a}^{(t)}, b^{(t)}) \big| s_0 = s \right], \tag{3}$$

---

[1]To ease the notation, and without any essential loss of generality, we will assume throughout that the action space does not depend on the state.

[2]Assuming that the reward is positive is without any loss of generality (see Claim D.6).

where the expectation is taken over the trajectory distribution induced by $\boldsymbol{\pi}_{\text{team}}$ and $\boldsymbol{\pi}_{\text{adv}}$. When the initial state is drawn from a distribution $\boldsymbol{\rho}$, the value function takes the form $V_{\boldsymbol{\rho}}(\boldsymbol{\pi}_{\text{team}}, \boldsymbol{\pi}_{\text{adv}}) \coloneqq \mathbb{E}_{s \sim \boldsymbol{\rho}}\Big[V_s(\boldsymbol{\pi}_{\text{team}}, \boldsymbol{\pi}_{\text{adv}})\Big]$.

**Nash Equilibrium.** Our main goal is to compute a joint policy profile that is an (approximate) *Nash equilibrium*, a standard equilibrium concept in game theory formalized below.

**Definition 2.1** (Nash equilibrium). *A joint policy profile* $\left(\boldsymbol{\pi}_{team}^{\star}, \boldsymbol{\pi}_{adv}^{\star}\right) \in \Pi$ *is an $\varepsilon$-approximate Nash equilibrium, for $\epsilon \geq 0$, if*

$$\begin{cases} V_{\boldsymbol{\rho}}(\boldsymbol{\pi}_{team}^{\star}, \boldsymbol{\pi}_{adv}^{\star}) \leq V_{\boldsymbol{\rho}}((\boldsymbol{\pi}_k', \boldsymbol{\pi}_{-k}^{\star}), \boldsymbol{\pi}_{adv}^{\star}) + \varepsilon, & \forall k \in [n], \forall \boldsymbol{\pi}_k' \in \Pi_k, \\ V_{\boldsymbol{\rho}}(\boldsymbol{\pi}_{team}^{\star}, \boldsymbol{\pi}_{adv}^{\star}) \geq V_{\boldsymbol{\rho}}(\boldsymbol{\pi}_{team}^{\star}, \boldsymbol{\pi}_{adv}') - \varepsilon, & \forall \boldsymbol{\pi}_{adv}' \in \Pi_{adv}. \end{cases}$$

That is, a joint policy profile is an (approximate) Nash equilibrium if no unilateral deviation from a player can result in a non-negligible—more than additive $\epsilon$—improvement for that player. Nash equilibria always exist in multi-agent stochastic games (Fink, 1964); our main result implies an (efficient) constructive proof of that fact for the special case of adversarial team Markov games.

### 2.3 Adversarial Markov Potential Games

A recent line of work has extended the fundamental class of potential normal-form games (Monderer & Shapley, 1996) to *Markov potential games* (Marden, 2012; Macua et al., 2018; Leonardos et al., 2021; Ding et al., 2022; Zhang et al., 2021b; Chen et al., 2022; Maheshwari et al., 2022; Fox et al., 2022). Importantly, our results readily carry over even if players in the team are not necessarily identically interested, but instead, there is some underlying potential function for the team; we will refer to such games as *adversarial Markov potential games*, formally introduced below.

**Definition 2.2.** *An adversarial Markov potential game $\mathcal{G} = (\mathcal{S}, \mathcal{N}, \mathcal{A}, \mathcal{B}, \{r_k\}_{k \in [n]}, \mathbb{P}, \gamma, \rho)$ is a multi-agent discounted MDP that shares all the properties of adversarial team Markov games (Section 2.1), with the exception that (2) is relaxed in that there exists a potential function $\Phi_s$, $\forall s \in \mathcal{S}$, such that for any $\boldsymbol{\pi}_{adv} \in \Pi_{adv}$,*

$$\Phi_s(\boldsymbol{\pi}_k, \boldsymbol{\pi}_{-k}; \boldsymbol{\pi}_{adv}) - \Phi_s(\boldsymbol{\pi}_k', \boldsymbol{\pi}_{-k}; \boldsymbol{\pi}_{adv}) = V_{k,s}(\boldsymbol{\pi}_k, \boldsymbol{\pi}_{-k}; \boldsymbol{\pi}_{adv}) - V_{k,s}(\boldsymbol{\pi}_k', \boldsymbol{\pi}_{-k}; \boldsymbol{\pi}_{adv}),$$

*for every agent $k \in [n]$, every state $s \in \mathcal{S}$, and all policies $\boldsymbol{\pi}_k, \boldsymbol{\pi}_{k'} \in \Pi_k$ and $\boldsymbol{\pi}_{-k} \in \Pi_{-k}$.*

## 3 Main Result

In this section, we sketch the main pieces required in the proof of our main result, Theorem 1.1. We begin by describing our algorithm in Section 3.1. Next, in Section 3.2, we characterize the strategy $\hat{\boldsymbol{x}} \in \mathcal{X}$ for the team returned by IPGMAX, while Section 3.3 completes the proof by establishing that $\hat{\boldsymbol{x}}$ can be efficiently extended to an approximate Nash equilibrium. The formal proof of Theorem 1.1 is deferred to the Appendix.

### 3.1 Our Algorithm

In this subsection, we describe in detail our algorithm for computing $\epsilon$-approximate Nash equilibria, IPGMAX, in adversarial team Markov games (Algorithm 1). IPGMAX takes as input a precision parameter $\epsilon > 0$ (Line 1) and an initial strategy for the team $(\boldsymbol{x}_1^{(0)}, \ldots, \boldsymbol{x}_n^{(0)}) = \boldsymbol{x}^{(0)} \in \mathcal{X} \coloneqq \bigtimes_{k=1}^{n} \mathcal{X}_k$ (Line 2). The algorithm then proceeds in two phases:

- In the first phase the team players are performing independent policy gradient steps (Line 7) with learning rate $\eta$, as defined in Line 3, while the adversary is then best responding to their joint strategy (Line 6). Both of these steps can be performed in polynomial time under oracle access to the game (see Remark 2). This process is repeated for $T$ iterations, with $T$ as defined in Line 4. We note that $\text{Proj}(\cdot)$ in Line 7 stands for the Euclidean projection, ensuring that each player selects a valid strategy. The first phase is completed in Line 9, where we set $\hat{\boldsymbol{x}}$ according to the iterate at time $t^{\star}$, for some $0 \leq t^{\star} \leq T - 1$. As we explain in Section 3.2, selecting uniformly at random is a practical and theoretically sound way of setting $t^{\star}$.

- In the second phase we are fixing the strategy of the team $\hat{\boldsymbol{x}} \in \mathcal{X}$, and the main goal is to determine a strategy $\hat{\boldsymbol{y}} \in \mathcal{Y}$ so that $(\hat{\boldsymbol{x}}, \hat{\boldsymbol{y}})$ is an $O(\epsilon)$-approximate Nash equilibrium. This is accomplished in the subroutine $\texttt{AdvNashPolicy}(\hat{\boldsymbol{x}})$, which consists of solving a linear program—from the perspective of the adversary—that has polynomial size. Our analysis of the second phase of IPGMAX can be found in Section 3.3.

It is worth stressing that under gradient feedback, IPGMAX requires no communication or coordination between the players in the team.

---

**Algorithm 1** Independent Policy GradientMax (IPGMAX)

---

1: Precision $\epsilon > 0$
2: Initial Strategy $\boldsymbol{x}^{(0)} \in \mathcal{X}$
3: Learning rate $\eta := \frac{\epsilon^2(1-\gamma)^9}{32S^4 D^2\left(\sum_{k=1}^n A_k + B\right)^3}$
4: Number of iterations $T := \frac{512 S^8 D^4 \left(\sum_{k=1}^n A_k + B\right)^4}{\epsilon^4(1-\gamma)^{12}}$
5: **for** $t \leftarrow 1, 2, \ldots, T$ **do**
6: $\quad \boldsymbol{y}^{(t)} \leftarrow \arg\max_{\boldsymbol{y} \in \mathcal{Y}} V_{\boldsymbol{\rho}}\left(\boldsymbol{x}^{(t-1)}, \boldsymbol{y}\right)$
7: $\quad \boldsymbol{x}_k^{(t)} \leftarrow \text{Proj}_{\mathcal{X}_k}\left(\boldsymbol{x}_k^{(t-1)} - \eta \nabla_{\boldsymbol{x}_k} V_{\boldsymbol{\rho}}\left(\boldsymbol{x}^{(t-1)}, \boldsymbol{y}^{(t)}\right)\right)$ $\qquad \triangleright$ for all agents $i \in [n]$
8: **end for**
9: $\hat{\boldsymbol{x}} \leftarrow \boldsymbol{x}^{(t^\star)}$
10: $\hat{\boldsymbol{y}} \leftarrow \texttt{AdvNashPolicy}(\hat{\boldsymbol{x}})$ $\qquad \triangleright$ defined in Algorithm 2
11: **return** $(\hat{\boldsymbol{x}}, \hat{\boldsymbol{y}})$

---

### 3.2 ANALYZING INDEPENDENT POLICY GRADIENTMAX

In this subsection, we establish that IPGMAX finds an $\epsilon$-nearly stationary point $\hat{\boldsymbol{x}}$ of $\phi(\boldsymbol{x}) := \max_{\boldsymbol{y} \in \mathcal{Y}} V_{\boldsymbol{\rho}}(\boldsymbol{x}, \boldsymbol{y})$ in a number of iterations $T$ that is polynomial in the natural parameters of the game, as well as $1/\epsilon$; this is formalized in Proposition 3.1.

First, we note the by-now standard property that the value function $V_{\boldsymbol{\rho}}$ is $L$-Lipschitz continuous and $\ell$-smooth, where $L := \frac{\sqrt{\sum_{k=1}^n A_k + B}}{(1-\gamma)^2}$ and $\ell := \frac{2\left(\sum_{k=1}^n A_k + B\right)}{(1-\gamma)^3}$ (Lemma C.1). An important observation for the analysis is that IPGMAX is essentially performing gradient descent steps on $\phi(\boldsymbol{x})$. However, the challenge is that $\phi(\boldsymbol{x})$ is not necessarily differentiable; thus, our analysis relies on the *Moreau envelope* of $\phi$, defined as follows.

**Definition 3.1** (Moreau Envelope). *Let* $\phi(\boldsymbol{x}) := \max_{\boldsymbol{y} \in \mathcal{Y}} V_{\boldsymbol{\rho}}(\boldsymbol{x}, \boldsymbol{y})$. *For any* $0 < \lambda < \frac{1}{\ell}$ *the Moreau envelope* $\phi_\lambda$ *of* $\phi$ *is defined as*

$$\phi_\lambda(\boldsymbol{x}) := \min_{\boldsymbol{x}' \in \mathcal{X}} \left\{ \phi(\boldsymbol{x}') + \frac{1}{2\lambda} \left\| \boldsymbol{x} - \boldsymbol{x}' \right\|^2 \right\}. \tag{4}$$

*We will let* $\lambda := \frac{1}{2\ell}$.

Crucially, the Moreau envelope $\phi_\lambda$, as introduced in (4), is $\ell$-strongly convex; this follows immediately from the fact that $\phi(\boldsymbol{x})$ is $\ell$-*weakly convex*, in the sense that $\phi(\boldsymbol{x}) + \frac{\ell}{2}\|\boldsymbol{x}\|^2$ is convex (see Lemma A.1). A related notion that will be useful to measure the progress of IPGMAX is the *proximal mapping* of a function $f$, defined as $\text{prox}_f : \mathcal{X} \ni \boldsymbol{x} \mapsto \arg\min_{\boldsymbol{x}' \in \mathcal{X}} \left\{ f(\boldsymbol{x}') + \frac{1}{2}\|\boldsymbol{x}' - \boldsymbol{x}\|^2 \right\}$; the proximal point of $\phi/(2\ell)$ is well-defined since $\phi$ is $\ell$-weakly convex (Proposition A.1). We are now ready to state the convergence guarantee of IPGMAX.

**Proposition 3.1.** Consider any $\epsilon > 0$. If $\eta = 2\epsilon^2(1-\gamma)$ and $T = \frac{(1-\gamma)^4}{8\epsilon^4(\sum_{k=1}^n A_k + B)^2}$, there exists an iterate $t^\star$, with $0 \leq t^\star \leq T - 1$, such that $\left\| \boldsymbol{x}^{(t^\star)} - \tilde{\boldsymbol{x}}^{(t^\star)} \right\|_2 \leq \epsilon$, where $\tilde{\boldsymbol{x}}^{(t^\star)} := \text{prox}_{\phi/(2\ell)}(\boldsymbol{x}^{(t^\star)})$.

The proof relies on the techniques of Lin et al. (2020), and it is deferred to Appendix C. The main takeaway is that $O(1/\epsilon^4)$ iterations suffice in order to reach an $\epsilon$-nearly stationary point of $\phi$—in the sense that it is $\epsilon$-far in $\ell_2$ distance from its proximal point. A delicate issue here is that Proposition 3.1 only gives a best-iterate guarantee, and identifying that iterate might introduce a

substantial computational overhead. To address this, we also show in Corollary C.1 that by randomly selecting $\lceil \log(1/\delta) \rceil$ iterates over the $T$ repetitions of IPGMAX, we are guaranteed to recover an $\epsilon$-nearly stationary point with probability at least $1 - \delta$, for any $\delta > 0$.

## 3.3 EFFICIENT EXTENSION TO NASH EQUILIBRIA

In this subsection, we establish that any $\epsilon$-nearly stationary point $\hat{x}$ of $\phi$, can be *extended* to an $O(\epsilon)$-approximate Nash equilibrium $(\hat{x}, \hat{y})$ for any adversarial team Markov game, where $\hat{y} \in \mathcal{Y}$ is the strategy for the adversary. Further, we show that $\hat{y}$ can be computed in polynomial time through a carefully constructed linear program. This "extendibility" argument significantly extends a seminal characterization of Von Stengel & Koller (1997), and it is the crux in the analysis towards establishing our main result, Theorem 1.1.

To this end, the techniques we leverage are more involved compared to (Von Stengel & Koller, 1997), and revolve around nonlinear programming. Specifically, in the spirit of (Filar & Vrieze, 2012, Chapter 3), the starting point of our argument is the following nonlinear program with variables $(\boldsymbol{x}, \boldsymbol{v}) \in \mathcal{X} \times \mathbb{R}^S$:

$$\min \ \sum_{s \in \mathcal{S}} \rho(s) v(s) + \ell \|\boldsymbol{x} - \hat{\boldsymbol{x}}\|^2$$

$$\text{s.t. } r(s, \boldsymbol{x}, b) + \gamma \sum_{s' \in \mathcal{S}} \mathbb{P}(s'|s, \boldsymbol{x}, b) v(s') \leq v(s), \quad \forall (s, b) \in \mathcal{S} \times \mathcal{B}; \qquad (Q1)$$

(Q-NLP)
$$\boldsymbol{x}_{k,s}^\top \mathbf{1} = 1, \quad \forall (k, s) \in [n] \times \mathcal{S}; \text{ and} \qquad (Q2)$$

$$x_{k,s,a} \geq 0, \quad \forall k \in [n], (s, a) \in \mathcal{S} \times \mathcal{A}_k. \quad (Q3)$$

Here, we have overloaded notation so that $r(s, \boldsymbol{x}, b) \coloneqq \mathbb{E}_{\boldsymbol{a} \sim \boldsymbol{x}_s}[r(s, \boldsymbol{a}, b]$ and $\mathbb{P}(s'|s, \boldsymbol{x}, b)) \coloneqq \mathbb{E}_{\boldsymbol{a} \sim \boldsymbol{x}_s}[\mathbb{P}(s'|s, \boldsymbol{a}, b)]$. For a fixed strategy $\boldsymbol{x} \in \mathcal{X}$ for the team, this program describes the (discounted) MDP faced by the adversary. A central challenge in this formulation lies in the nonconvexity-nonconcavity of the constraint functions, witnessed by the multilinear constraint ($Q1$). Importantly, unlike standard MDP formulations, we have incorporated a quadratic regularizer in the objective function; this term ensures the following property.

**Proposition 3.2.** For any fixed $\boldsymbol{x} \in \mathcal{X}$, there is a unique optimal solution $\boldsymbol{v}^\star$ to (Q-NLP). Further, if $\tilde{\boldsymbol{x}} \coloneqq \text{prox}_{\phi/(2\ell)}(\hat{\boldsymbol{x}})$ and $\tilde{\boldsymbol{v}} \in \mathbb{R}^S$ is the corresponding optimal, then $(\tilde{\boldsymbol{x}}, \tilde{\boldsymbol{v}})$ is the global optimum of (Q-NLP).

The uniqueness of the associated value vector is a consequence of Bellman's optimality equation, while the optimality of the proximal point follows by realizing that (Q-NLP) is an equivalent formulation of the proximal mapping. These steps are formalized in Appendix B.2. Having established the optimality of $(\tilde{\boldsymbol{x}}, \tilde{\boldsymbol{v}})$, the next step is to show the existence of nonnegative Lagrange multipliers satisfying the KKT conditions (recall Definition A.2); this is non-trivial due to the nonconvexity of the feasibility set of (Q-NLP).

To do so, we leverage the so-called *Arrow-Hurwicz-Uzawa constraint qualification* (Theorem A.1)—a form of "regularity condition" for a nonconvex program. Indeed, in Lemma B.3 we show that any feasible point of (Q-NLP) satisfies that constraint qualification, thereby implying the existence of nonnegative Lagrange multipliers satisfying the KKT conditions for any local optimum (Corollary B.1), and in particular for $(\tilde{\boldsymbol{x}}, \tilde{\boldsymbol{v}})$:

**Proposition 3.3.** There exist nonnegative Lagrange multipliers satisfying the KKT conditions at $(\tilde{\boldsymbol{x}}, \tilde{\boldsymbol{v}})$.

Now the upshot is that a subset of those Lagrange multipliers $\tilde{\boldsymbol{\lambda}} \in \mathbb{R}^{S \times B}$ can be used to establish the extendibility of $\hat{x}$ to a Nash equilibrium. Indeed, our next step makes this explicit: We construct a linear program whose sole goal is to identify such multipliers, which in turn will allow us to efficiently compute an admissible strategy for the adversary $\hat{y}$. However, determining $\tilde{\boldsymbol{\lambda}}$ exactly seems too ambitious. For one, IPGMAX only granted us access to $\hat{x}$, but not to $\tilde{x}$. On the other hand, the Lagrange multipliers $\tilde{\boldsymbol{\lambda}}$ are induced by $(\tilde{\boldsymbol{x}}, \tilde{\boldsymbol{v}})$. To address this, the constraints of our linear program are phrased in terms of $(\hat{\boldsymbol{x}}, \hat{\boldsymbol{v}})$, instead of $(\tilde{\boldsymbol{x}}, \tilde{\boldsymbol{v}})$, while to guarantee feasibility we

appropriately relax all the constraints of the linear program; this relaxation does not introduce a large error since $\|\hat{x} - \tilde{x}\| \leq \epsilon$ (Proposition 3.1), and the underlying constraint functions are Lipschitz continuous—with constants that depend favorably on the game $\mathcal{G}$; we formalize that in Lemma B.4. This leads to our main theorem, summarized below (see Theorem B.1 for a precise statement).

**Theorem 3.1.** *Let $\hat{x}$ be an $\epsilon$-nearly stationary point of $\phi$. There exist a linear program, (LP$_{adv}$), such that:*

*(i) It has size that is polynomial in $\mathcal{G}$, and all the coefficients depend on the (single-agent) MDP faced by the adversary when the team is playing a fixed strategy $\hat{x}$; and*

*(ii) It is always feasible, and any solution induces a strategy $\hat{y}$ such that $(\hat{x}, \hat{y})$ is an $O(\epsilon)$-approximate Nash equilibrium.*

The proof of this theorem carefully leverages the structure of adversarial team Markov games, along with the KKT conditions we previously established in Proposition 3.3. The algorithm for computing the policy for the adversary is summarized in Algorithm 2 of Appendix B. A delicate issue with Theorem 3.1, and in particular with the solution of (LP$_{adv}$), is whether one can indeed *efficiently simulate* the environment faced by the adversary. Indeed, in the absence of any structure, determining the coefficients of the linear program could scale exponentially with the number of players; this is related to a well-known issue in computational game theory, revolving around the exponential blow-up of the input space as the number of players increases (Papadimitriou & Roughgarden, 2008). As is standard, we bypass this by assuming access to natural oracles that ensure we can efficiently simulate the environment faced by the adversary (Remark 2).

## 4 FURTHER RELATED WORK

In this section, we highlight certain key lines of work that relate to our results in the context of adversarial team Markov games. We stress that the related literature on multi-agent reinforcement learning (MARL) is too vast to even attempt to faithfully cover here. For some excellent recent overviews of the area, we refer the interested reader to (Yang & Wang, 2020; Zhang et al., 2021a) and the extensive lists of references therein.

**Team Games.** The study of team games has been a prolific topic of research in economic theory and group decision theory for many decades; see, *e.g.*, (Marschak, 1955; Groves, 1973; Radner, 1962; Ho & Chu, 1972). A more modern key reference point to our work is the seminal paper of Von Stengel & Koller (1997) that introduced the notion of *team-maxmin equilibrium (TME)* in the context of normal-form games. A TME profile is a mixed strategy for each team member so that the minimal expected team payoff over all possible responses of the adversary—who potentially knows the play of the team—is the maximum possible. While TME's enjoy a number of compelling properties, being the optimal equilibria for the team given the lack of coordination, they suffer from computational intractability even in 3-player team games (Hansen et al., 2008; Borgs et al., 2010).[3] Nevertheless, practical algorithms have been recently proposed and studied for computing them in multiplayer games (Zhang & An, 2020a;b; Basilico et al., 2017). It is worth pointing out that team equilibria are also useful for extensive-form two-player zero-sum games where one of the players has *imperfect recall* (Piccione & Rubinstein, 1997).

The intractability of TME has motivated the study of a relaxed equilibrium concept that incorporates a *correlation device* (Farina et al., 2018; Celli & Gatti, 2018; Basilico et al., 2017; Zhang & An, 2020b; Zhang & Sandholm, 2021; Zhang et al., 2022b; Carminati et al., 2022; Zhang et al., 2022a); namely, *TMECor*. In TMECor players are allowed to select *correlated strategies*. Despite the many compelling aspects of TMECor as a solution concept in team games, even *ex ante* coordination or correlated randomization—beyond the structure of the game itself—can be overly expensive or even infeasible in many applications (Von Stengel & Koller, 1997). Further, even TMECor is NP-hard to compute (in the worst-case) for *imperfect-information* extensive-form games (EFGs) (Chu & Halpern, 2001), although fixed-parameter-tractable (FPT) algorithms have recently emerged for natural classes of EFGs (Zhang & Sandholm, 2021; Zhang et al., 2022b).

---

[3]Hansen et al. (2008); Borgs et al. (2010) establish FNP-hardness and inapproximability for general 3-player games, but their argument readily applies to 3-player team games as well.

On the other hand, the computational aspects of the standard Nash equilibrium (NE) in adversarial team games is not well-understood, even in normal-form games. In fact, it is worth pointing out that Von Neumann's celebrated *minimax theorem* (von Neumann & Morgenstern, 2007) does not apply in team games, rendering traditional techniques employed in two-player zero-sum games of little use. Indeed, Schulman & Vazirani (2017) provided a precise characterization of the *duality gap* between the two teams based on the natural parameters of the problem, while Kalogiannis et al. (2021) showed that standard no-regret learning dynamics such as gradient descent and optimistic Hedge could fail to stabilize to mixed NE even in binary-action adversarial team games. Finally, we should also point out that although from a complexity-theoretic standpoint our main result (Theorem 1.1) establishes a *fully polynomial time approximate scheme (FPTAS)*, since the dependence on the approximation error $\epsilon$ is poly($1/\epsilon$), an improvement to poly($\log(1/\epsilon)$) is precluded even in normal-form games unless CLS $\subseteq$ P (an unlikely event); this follows as adversarial team games capture potential games (Kalogiannis et al., 2021), wherein computing mixed Nash equilibria is known to be complete for the class CLS = PPAD $\cap$ PLS (Babichenko & Rubinstein, 2021).

**Multi-agent RL.** Computing Nash equilibria has been a central endeavor in multi-agent RL. While some algorithms have been proposed, perhaps most notably the Nash-Q algorithm (Hu & Wellman, 1998; 2003), convergence to Nash equilibria is only guaranteed under severe restrictions on the game. More broadly, the long-term behavior of independent policy gradient methods (Schulman et al., 2015) is still not well-understood. Before all else, from the impossibility result of Hart & Mas-Colell, universal convergence to Nash equilibria is precluded even for normal-form games; this is aligned with the computational intractability (PPAD-completeness) of Nash equilibria even in two-player general-sum games (Daskalakis et al., 2009; Chen et al., 2009). Surprisingly, recent work has also established hardness results in turn-based stochastic games, rendering even the weaker notion of (stationary) CCEs intractable (Daskalakis et al., 2022; Jin et al., 2022).

As a result, the existing literature has inevitably focused on specific classes of games, such as Markov potential games (Leonardos et al., 2021; Ding et al., 2022; Zhang et al., 2021b; Chen et al., 2022; Maheshwari et al., 2022; Fox et al., 2022) or two-player zero-sum Markov games (Daskalakis et al., 2020; Wei et al., 2021; Sayin et al., 2021; Cen et al., 2021; Sayin et al., 2020). As we pointed out earlier, adversarial Markov team games can unify and extend those settings (Section 2.3). More broadly, identifying multi-agent settings for which Nash equilibria are provably efficiently computable is recognized as an important open problem in the literature (see, *e.g.*, (Daskalakis et al., 2020)), boiling down to one of the main research question of this paper (Question (★)). We also remark that certain guarantees for convergence to Nash equilibria have been recently obtained in a class of symmetric games (Emmons et al., 2022)—including symmetric team games. Finally, weaker solution concepts relaxing either the Markovian or the stationarity properties have also recently attracted attention (Daskalakis et al., 2022; Jin et al., 2021).

## 5 CONCLUSIONS

Our main contribution in this paper is the first polynomial algorithm for computing (stationary) Nash equilibria in adversarial team Markov games, an important class of games in which a team of uncoordinated but identically-interested players is competing against an adversarial player. We argued that this setting serves as a step towards modeling more realistic multi-agent applications that feature both competing and cooperative interests.

There are many interesting directions for future research. One caveat of our main algorithm (IPGMAX) is that it requires a separate subroutine for computing the optimal policy of the adversary. It is plausible that a carefully designed two-timescale policy gradient method can efficiently reach a Nash equilibrium, which would yield fully model-free algorithms for adversarial team Markov games by obviating the need to solve a linear program. Techniques from the literature on constrained MDPs (Ying et al., 2022) could also be useful for computing the policy of the adversary in a more scalable way. Furthermore, exploring different solution concepts—beyond Nash equilibria—could also be a fruitful avenue for the future. Indeed, allowing some limited form of correlation between the players in the team could lead to more efficient algorithms; whether that form of coordination is justified (arguably) depends to a large extent on the application at hand. Finally, returning to Question (★), a more ambitious agenda revolves around understanding the fundamental structure of games for which computing Nash equilibria is provably computationally tractable.

## ACKNOWLEDGMENTS

We are grateful to the anonymous ICLR reviewers for their valuable feedback. Ioannis Anagnostides thanks Gabriele Farina and Brian H. Zhang for helpful discussions. Ioannis Panageas would like to acknowledge a start-up grant. Part of this project was done while he was a visiting research scientist at the Simons Institute for the Theory of Computing for the program "Learning and Games". Vaggos Chatziafratis was supported by a start-up grant of UC Santa Cruz, the Foundations of Data Science Institute (FODSI) fellowship at MIT and Northeastern, and part of this work was carried out at the Simons Institute for the Theory of Computing. Emmanouil V. Vlatakis-Gkaragkounis is grateful for financial support by the Google-Simons Fellowship, Pancretan Association of America and Simons Collaboration on Algorithms and Geometry. This project was completed while he was a visiting research fellow at the Simons Institute for the Theory of Computing. Additionally, he would like to acknowledge the following series of NSF-CCF grants under the numbers 1763970/2107187/1563155/1814873.

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
