# OpenReview forum: "Efficiently Computing Nash Equilibria in Adversarial Team Markov Games"
_ICLR.cc/2023/Conference — ICLR 2023 notable top 5%_

### Official Review · Reviewer_V4b6 · 2022-10-20

**Confidence:** 3
**Clarity, Quality, Novelty And Reproducibility:** I think there are some problems about…
**Correctness:** 4
**Technical Novelty And Significance:** 3
**Empirical Novelty And Significance:** Not applicable
**Recommendation:** 8

**Strength And Weaknesses:**

Strength

Looking for Nash Equilibrias in Markov Game is an important task, and reducing the time complexity from exponential to polynomial is also a significant contribution.

Weaknesses

There are many parts not very clear in the IPGMAX algorithm (for details please see the questions below)

My Questions

1. Proposition 3.1 shows that there must exists such a $t^*$, and Corollary C.1 shows that with high probability, there is one such $t^*$ in the randomly chosen set. However, it is still not very clear to me that how you can find this $t^*$ is this set. Do you mean that it is easy (with low computation cost) to check whether $x^{t}$ is good or not? How much is the computation cost here?

2. In line 6 of Algorithm 1, we need to look for the best response of the adversary. What is the computation cost here?

3. In line 7 of Algorithm 1, we need to compute the gradient of $V_{\rho}$. What is the computation cost here?

4. It is mentioned that the IPGMAX algorithm is a decentralized one (which may avoid communications between players), but I am wondering whether line 6 and line 7 in Algorithm 1 could be done without any communications? Do you mean that these steps could be done even if all the players (e.g. the team players) do not know others' policies in the last time step?





**Summary Of The Paper:**

This paper provides an efficient algorithm to compute the Nash Equilibria in an adversarial team Markov game. Specifially, in this zero sum Markov game, a team of players try to gain reward as much as possible, and an adversary tries to lose reward as less as possible. The authors propose an algorithm called IPGMAX. In this algorithm, all the team players will first do policy gradient descent for several steps, and this procedure returns the team players' policies in Nash Equilibria. Then the algorithm uses a procedure to look for the response of the adversarial policy (to the team players' policies). The authors show that, team players' policies as well as the the response of the adversarial policy form a Nash Equilibria, and the time cost of the algorithm is polynomial, which can be much better than the exponential ones in prior works.

**Summary Of The Review:**

Overall, I have some questions about the algorithm in this paper, and give the score of "weak accept". I'd like to see the answers from the authors and I will change my score if the answers are convincing.

=========After Rebuttal=========

The rebuttal explains about the details, and now I can understand how the algorithm works. I believe those details should also be included in the final version, so that the readers can easily understand it as well.

My remaining concerns (which are also mentioned by the other reviewers) are about the existance of the oracle and whether the given complexity upper bound is tight. I think some further works about the tightness could be done in the future.

Overall, I agree that this work is an important step in efficiently looking for the Nash Equilibria in Marcov Games, and I would like change my score to "accept".

---

> ### Author Response · Authors · 2022-11-07
> **Re: Official Review of Paper3334 by Reviewer V4b6**
>
>
>
> We thank the reviewer for the helpful feedback. Below we address the reviewer's questions.
>
> >- "*Proposition 3.1 shows that there must exists such a $t^\*$,
> and Corollary C.1 shows that with high probability, there is one such in the randomly chosen set. However, it is still not very clear to me that how you can find this is this set. Do you mean that it is easy (with low computation cost) to check whether $x^t$ is good or not? How much is the computation cost here?*"
>
> We can indeed efficiently check whether $x^t$ is "good" by deriving the corresponding policy of the adversary $y^t$ (Line 10 of IPGMax), and then determining the Nash equilibrium gap of $(x^t, y^t)$, which can be computed efficiently (that is, in polynomial time) by solving single-agent MDPs. Corollary C.1 guarantees that a number of $\mathrm{log} (1/\delta)$ (random) samples/repetitions suffice with probability at least $1 - \delta$, so this only leads to a logarithmic overhead.
>
>
> >- "*In line 6 of Algorithm 1, we need to look for the best response of the adversary. What is the computation cost here?*"
>
> The computation here boils down to the solution of a single-agent MDP, for which by now there are numerous efficient algorithms, *e.g.*, Q-learning [6] or linear programming [7].
>
>
>
> >- "*In line 7 of Algorithm 1, we need to compute the gradient of $V_\rho$. What is the computation cost here?*"
>
> It is common to assume oracle-access to the gradients of the value function in the theory of policy gradient methods ([1]). This assumption can be justified as follows: We can compute the gradient efficiently by computing the $Q$-function of every agent $k$ and their discounted state visitation measure $d_{\rho}$ given a policy $x_k$:
>
> $$ \frac{\partial }{\partial x_{k,s,a} } V_\rho(x,y)=  d_{\rho}^{x,y} (s) Q^{x,y}(s,a). \quad \quad \quad (1)$$
>
> The $Q$-function can be computed after obtaining the value function $V_\rho$, which in turn is computed by solving a linear program in polynomial time [4, 5]. Also, $d_{\rho}$--the state visitation measure---can be computed in polynomial time through linear programming [5; Theorem 6.9.1].
>
> It is also worth mentioning that using [2; Theorem 31] and [3] it readily follows that having an estimate of the gradient suffices to show convergence to a near stationary point, so we don't necessarily need exact gradients.
>
>
>
> >- "*It is mentioned that the IPGMAX algorithm is a decentralized one (which may avoid communications between players), but I am wondering whether line 6 and line 7 in Algorithm 1 could be done without any communications? Do you mean that these steps could be done even if all the players (e.g. the team players) do not know others' policies in the last time step?*"
>
> Our algorithm is indeed decentralized under the standard and unavoidable assumption that each player receives suitable feedback from the environment during the execution of the algorithm. In particular,
>
> * Line 6 requires solving a single-agent MDP, a best-response problem from the side of the adversary whereby every team member $k$ has commited to a **fixed** strategy $x_k$. This step requires no communication between the players, and can be implemented in a decentralized fashion under minimal and standard assumptions for the feedback received by the adversary.
>
> * Line 7 is also fully decentralized and requires no communication between team members under the standard assumption that each player receives gradient feedback.
>
> <br>
>
>
> [1] Agarwal, A., Kakade, S.M., Lee, J.D. and Mahajan, G., 2021. On the Theory of Policy Gradient Methods: Optimality, Approximation, and Distribution Shift. J. Mach. Learn. Res., 22(98), pp.1-76.
>
> [2] Jin, C., Netrapalli, P. and Jordan, M., 2020, November. What is local optimality in nonconvex-nonconcave minimax optimization?. In International conference on machine learning (pp. 4880-4889). PMLR.
>
> [3] Damek Davis and Dmitriy Drusvyatskiy. Stochastic subgradient method converges at the rate o(k^{-1/4} ) on
> weakly convex functions. arXiv preprint arXiv:1802.02988, 2018
>
> [4] Yinyu Ye. The simplex and policy-iteration methods are strongly polynomial for the markov decision problem with a
> fixed discount rate. Math. Oper. Res., 36(4):593–603, 2011.
>
> [5] Puterman, M.L., 2014. Markov decision processes: discrete stochastic dynamic programming. John Wiley & Sons.
>
>
> [6] Jin, C., Allen-Zhu, Z., Bubeck, S. and Jordan, M.I., 2018. Is Q-learning provably efficient?. Advances in neural information processing systems, 31.
>
> [7] Yinyu Ye. The simplex and policy-iteration methods are strongly polynomial for the markov decision problem with a fixed discount rate. Math. Oper. Res., 36(4):593–603, 2011.

---

> > ### Comment · Reviewer_V4b6 · 2022-11-07
> > **Further Questions**
> >
> > Thanks for your answers!
> >
> > There are still some points not very clear to me.
> >
> > For the following answer:
> >
> > "In line 6 of Algorithm 1, we need to look for the best response of the adversary. What is the computation cost here?"
> > The computation here boils down to the solution of a single-agent MDP, for which by now there are numerous efficient algorithms, e.g., Q-learning [6] or linear programming [7].
> >
> > Do you mean that in this line, we need all the team agents fix their policy, and let the adversary run an RL policy to look for his best response? In this case, how can the decentralized players synchronize with each other?
> >
> > Similarly, in line 7 of Algorithm 1, do we need to let all the agents run their policies for several time steps and then estimate the corresponding gradients?

---

> > > ### Author Response · Authors · 2022-11-08
> > > **Re: Further Questions of Reviewer V4b6**
> > >
> > > We thank the reviewer for following-up.
> > >
> > >
> > > >- Do you mean that in this line, we need all the team agents fix their policy, and let the adversary run an RL policy to look for his best response? In this case, how can the decentralized players synchronize with each other?
> > >
> > > This is correct: as part of Line 6 every team agent fixes their policy and then the adversary has to compute a best response under a stationary environment. This reduces to a single-agent MDP, and can be implemented in a decentralized fashion as long as the adversary receives feedback (e.g. the expected reward under a given action) from their environment for a sufficient number of repetitions--having fixed the policies of all team players. Each team agent gets feedback for their own values and update their own policy independently from the others per iterate, this leads to a decentralized algorithm with running time that scales with the individual action spaces and **not the joint one**.
> > >
> > > Decentralization has to do with the feedback (value and gradient) each agent gets.
> > >
> > >
> > > >- Similarly, in line 7 of Algorithm 1, do we need to let all the agents run their policies for several time steps and then estimate the corresponding gradients?
> > >
> > > Assuming gradient-oracle access, it suffices to have each player perform a *single* gradient descent step. So there is no need to run their policies for several time steps. But even without gradient-oracle, we could have each player *estimate* the gradient by running their policies for several time steps, as the reviewer points out.
> > >
> > > In words, our algorithm alternates between a best-response step by the adversary (for fixed team agents) and simultaneous policy gradient updates by the team players (with the adversary remaining fixed).
> > >
> > > Please let us know if we have clarified the reviewer's questions, and if there are any further follow-up questions.

---

> > > > ### Comment · Reviewer_V4b6 · 2022-11-10
> > > > **Re: Re: Further Questions of Reviewer V4b6**
> > > >
> > > > In my opinion, Q-Learning or other single-agent RL algorithms only converge to the best response (of the adversary) when the number of time steps is infinite. However, if the number of time steps is finite (e.g., about $O({1\over \epsilon^2})$), then we can only obtain an $\epsilon$-approximate best response. Would this error of $\epsilon$ influence your theoretical results? For example, in Line 6 of Algorithm 1, would it matter if $y^{(t)}$ is only an $\epsilon$-approximate best response but not the exact best response?

---

> > > > > ### Author Response · Authors · 2022-11-11
> > > > > **Re: Re: Re: Further Questions of Reviewer V4b6 (approx. best-response)**
> > > > >
> > > > > This is a very good and reasonable question. Our main theoretical result directly applies even if $y^{(t)}$ is an $\epsilon'$-approximate best response, as long as $\epsilon'$ is small enough. More precisely, assuming that $y^{(t)}$ is an $\epsilon$-approximate best response only alters the proof of Proposition 3.1 in the following way: Eq. (41) now holds with an $\epsilon$ additive error, which in turn translates to a $2\epsilon' \eta \ell$ additive factor in Eq. (46). So, as long as $\epsilon' = O(\eta)$ the rest of the argument immediately applies.
> > > > >
> > > > > This means that one can employ RL algorithms that only compute an approximate best response in finite time, such as Q-learning. That being said, there are polynomial-time algorithms that compute an exact best response (see Ye (2011): The simplex and policy-iteration methods are strongly polynomial for the Markov decision problem with a fixed discount rate), although algorithms such as a Q-learning are arguably more scalable.
> > > > >
> > > > > We added a remark clarifying this point in the revised version and we kindly ask that you check the new version to make sure that we have made this point clear (Remark 1 & Remark 4).

---

> > > > > > ### Comment · Reviewer_V4b6 · 2022-11-14
> > > > > > **Thanks for the reply**
> > > > > >
> > > > > > Thanks for your detailed explanations. Now I can understand the whole procedure of your algorithms.

---

> > > > > > > ### Author Response · Authors · 2022-11-14
> > > > > > > **Thanks for the discussion**
> > > > > > >
> > > > > > > We are glad to have engaged in this discussion and to have been able to answer your questions all of which were very reasonable.
> > > > > > >
> > > > > > > Nevertheless, we would like to note that oracle-access to the gradients and/or the value functions should not be considered as a caveat or drawback of our algorithm. Rather, it is standard for the *full-information feedback* regime which is evident in a number of works, *e.g.,* [1],[2].
> > > > > > >
> > > > > > >
> > > > > > > Thanks,
> > > > > > >
> > > > > > > The authors
> > > > > > >
> > > > > > > [1]  Wei, C.Y., Lee, C.W., Zhang, M. and Luo, H., 2021, July. Last-iterate convergence of decentralized optimistic gradient descent/ascent in infinite-horizon competitive markov games. In Conference on learning theory (pp. 4259-4299). PMLR.
> > > > > > >
> > > > > > > [2] Zhang, R., Liu, Q., Wang, H., Xiong, C., Li, N. and Bai, Y., 2022. Policy Optimization for Markov Games: Unified Framework and Faster Convergence. arXiv preprint arXiv:2206.02640.

---

### Official Review · Reviewer_1bpS · 2022-10-24

**Confidence:** 2
**Correctness:** 4
**Technical Novelty And Significance:** 4
**Empirical Novelty And Significance:** Not applicable
**Recommendation:** 8

**Clarity, Quality, Novelty And Reproducibility:**

+ The paper is written very clearly and is well organized.
+ The novelty is put into context well and is clear.

**Strength And Weaknesses:**

# Strengths
- The paper introduces the first poly-time algorithm to compute a Nash equilibrium in the setting of adversarial team Markov games. Computing a Nash equilibrium is an important practical problem, but unfortunately the general-case is difficult. Therefore, it is useful and relevant to consider special cases such as the coalition considered here.
- The algorithm has good scaling with the action sets of each agent; that is, it scales with their sum rather than the product.
- The techniques used to obtain this result are well described at an overview level and compared to the most relevant related work which is to compute a Nash equilibrium in the analogous normal-form setting. The challenges in generalizing the normal-form setting to Markov game include: nonlinear program with a set of nonconvex constraints, which requires the Arrow-Hurwiz-Uzawa constraint qualification technique.

# Weaknesses
- The type of game considered is very special. It is only a modest generalization of two-player zero-sum games. More practical would be algorithms that have exponential worst-case but still run on practical examples of n-player Markov games. Still, any generalizations from fully cooperative or competitive two-player games are welcome.
- The practical scalability of the algorithm is not evaluated. Although it is polynomial, the number of iterations looks very large from inspection of the pseudocode.  Still, this is a theoretical work, so this isn't too significant of a weakness.



**Summary Of The Paper:**

This paper considers multi-agent Markov games; specifically, the problem of computing a Nash equilibrium. Even in normal-form games, it is computationally intractable to compute a Nash equilibrium in two-player general-sum games, much less in games with more players. In this paper, the game is restricted to coalition of players with a common adversary, so-called "adversarial team Markov games". In addition to the case when the coalition shares a common objective, the results extend to the case in which the objectives of the coalition share a common potential function (Markov potential games).

**Summary Of The Review:**

Based on the above strengths and weaknesses, the paper clearly is in the "Accept" category.

---

> ### Author Response · Authors · 2022-11-07
> **Re: Official Review of Paper3334 by Reviewer 1bpS**
>
> We thank the reviewer for the helpful feedback. Below we address the reviewer's main points.
>
> >- "*The type of game considered is very special. It is only a modest generalization of two-player zero-sum games. More practical would be algorithms that have exponential worst-case but still run on practical examples of n-player Markov games. Still, any generalizations from fully cooperative or competitive two-player games are welcome.*"
>
> While we agree with the reviewer in that team games are a rather special class of games, we argue that they are very relevant in many practical scenarios, which is part of the reason why they have received a tremendous amount of interest in the literature. Moreover, adversarial team Markov games unify two-player zero-sum Markov games and Markov potential games, so we consider them to be a substantial generalization of two-player zero-sum games. We view this unification as an important contribution of our work given the recent surge of interest in computing Nash equilibria in either of those two special cases. As we noted in our answer to Reviewer S2Qp, this class of games captures **both cooperation and competition**.
>
>
> >- "*The practical scalability of the algorithm is not evaluated. Although it is polynomial, the number of iterations looks very large from inspection of the pseudocode. Still, this is a theoretical work, so this isn't too significant of a weakness.*"
>
> This is a fair point. We did not attempt to optimize the iteration complexity in terms of the paramters of the game, although we agree with the reviewer that it is an important direction. Still, in terms of the dependence on $\epsilon$, our rates match the standard bounds needed to compute near stationary points of nonsmooth functions, which is an inherent barrier in our approach.

---

### Official Review · Reviewer_S2Qp · 2022-10-25

**Confidence:** 3
**Correctness:** 4
**Technical Novelty And Significance:** 2
**Empirical Novelty And Significance:** 2
**Recommendation:** 5

**Clarity, Quality, Novelty And Reproducibility:**

Clarity could be improved in some parts of the paper. The results look novel and very technical.

**Strength And Weaknesses:**

Strength:

The problem is well-motivated and the paper is written well overall. The proposed approach to computing an approximate Nash equilibrium looks non-trivial and brings in many interesting concepts. The authors also provide a good summary about the related work.

Weakness:

- Section 3.3 could have been improved to make the main idea clearer. It is unclear to me what is the main advantage of the proposed methods? In particular, why not just solve Q-NLP without the regularizer in the objective function, which gives a Nash equilibrium directly and seems much more manageable than the current formulation?

- The approach relies on an oracle to tackle a computational obstacle, which may be crucial. This further deepens the question of how meaningful the proposed methods are compared with solving Q-NLP without the regularizer --- now that there's an oracle to use, so supposedly it also simplifies the problem of solving Q-NLP without the regularizer.


**Summary Of The Paper:**

The paper studies a zero-sum team Markov game. In the game, a team of agents compete with an adversary. Agents in the team have the same reward function, and the sum of the team and the adversary's rewards is zero. The paper in particualr looks at a class of potential games. The main contribution of the paper is to propose a set of algorithms to compute a stationary epsilon-Nash equilibrium of the game.

**Summary Of The Review:**

A paper on a well-motivated problem, overall well written but with some issues on the clarifty. Results are comprehensive and very technical, but with some weaknesses that raise questions about how meaningful the results are, which may or may not be crucial.

---

> ### Author Response · Authors · 2022-11-07
> **Re: Official Review of Paper3334 by Reviewer S2Qp**
>
> We thank the reviewer for the helpful feedback. Below we address the concerns.
>
> >- "*The paper in particular looks at a class of potential games.*"
>
> To clarify, our paper does not *just* solve the problem of computing Nash equilibria in Markov potential games (settings encompassing cooperation), but tackles a **generalization** (of potential games) thereof whereby an adversarial agent controls the potential function. We stress that to the best of our knowledge, no prior algorithms (for approximating Nash equilibrium policies) that are polynomial in the parameters of the game were known for games that contain **both cooperation and competition**.
>
>
>
> >- "*Section 3.3 could have been improved to make the main idea clearer. It is unclear to me what is the main advantage of the proposed methods? In particular, why not just solve Q-NLP without the regularizer in the objective function, which gives a Nash equilibrium directly and seems much more manageable than the current formulation?*"
>
> The purpose of Section 3.3 is to extend an approximate stationary point $\hat{x}^t$ to an approximate Nash equilibrium $(\hat{x}^t, \hat{y}^t)$. We clarify the following key points:
>
> - It is by no means enough to compute a best response $\hat{y}^t$ for the adversary. For example, consider the two-player zero-sum game of rock-papers-scissors; if one player is mixing uniformly at random, then **any** strategy for the other player is a best response, but of course the induced pair of strategies could be very far from being a Nash equilibrium. So, to answer the reviewer's question, *solving Q-NLP without the regularizer is by no means guaranteed to give a Nash equilibrium*.
>
> - Further, notwithstanding the above remark, Q-NLP has a non-convex feasible region (in other words, the constraints are non-convex), and solving such programs in general is NP-hard.
>
> Instead, a more refined approach is needed. In particular, as we explain in Section 3.3, Q-NLP is never solved in our algorithm, but it is used as an intermediate auxiliary step towards proving the extendibility via a carefully constructed linear program. More precisely, the role of Q-NLP in our argument is to establish the existence of nonnegative Lagrange multipliers satisfying the KKT conditions (Proposition 3.3), and it turns out that the quadratic regularizer in Q-NLP is crucial for our proof of Proposition 3.3. In other words, we have not been able to establish the existence of nonnegative Lagrange multipliers **without** the quadratic regularizer. Next, leveraging Proposition 3.3, we were able to show that the policy of the adversary can be derived as the solution of a suitable linear program.
>
> This *extendibility argument is really the crux to our main result*. Even in the very special case of two-player zero-sum normal-form games, such an extendibility property is a consequence of the highly non-trivial minimax theorem; further, for the special case of adversarial team normal-form games, a much weaker extendibility result was the main contribution of the seminal work of Von Stengel and Koller (1997). So, the advantage of our proposed method is that it leads to the *first* polynomial time algorithm for computing Nash equilibria in adversarial team Markov games. We hope that this addresses the reviewer's concerns and clarifies the purpose of Section 3.3.
>
>
> >- "The approach relies on an oracle to tackle a computational obstacle, which may be crucial. This further deepens the question of how meaningful the proposed methods are compared with solving Q-NLP without the regularizer --- now that there's an oracle to use, so supposedly it also simplifies the problem of solving Q-NLP without the regularizer."
>
> As we explained above, there are computational barriers to solving Q-NLP (completely unrelated to the oracle we posit), but also its solution does not necessarily lead to a Nash equilibrium. Now with regards to the oracle we use, as we point out there is an inherent need for such an oracle in multiplayer games, for otherwise the size of the input would scale exponentially with the number of the players. This is an unavoidable and standard assumption in multiplayer games; typically one of the following is assumed: either that one has oracle access to the values of the value functions and their gradients *or* the game exhibits a succinct representation. Nonwithstanding the above point, our results also apply and are meaningful even when no such oracle is given; the complexity of our algorithm then would be exponential in the number of the players, but note that now this is polynomial in the size of the input---so the complexity of our algorithm is again polynomial. But we believe that it is more natural to work with the standard oracle we assumed in order to obtain a polynomial dependency in the number of players.

---

> > ### Author Response · Authors · 2022-11-14
> > **Further clarifications? Only a few days more for discussion.**
> >
> > We would like to ask whether you want further clarifications or whether we could address additional questions you might have. Please take into consideration that the discussion period has only a few days left.

---

### Decision · Program_Chairs · 2023-01-20

**Decision:**

Accept: notable-top-5%

**Justification For Why Not Higher Score:**

n/a

**Justification For Why Not Lower Score:**

Two reviewers were were very positive, and the content of the marginally-negative reviewer's was very similar.  The reviewers agreed that the approach was non-trivial and a significant improvement on the state of the art.  The problem is a special case, but not contrived.

**Metareview: Summary, Strengths And Weaknesses:**

(a) Summary: This paper proposes a technique by which a team of agents playing a zero-sum Markov game against a single adversary can compute a Nash equilibrium.  The procedure is asymptotically efficient (polynomial in the largest action set, rather than in the size of action profile set).

(b) Strengths: The reviewers agreed that the scaling of the algorithm is a significant improvement.  The algorithm is a non-trivial contribution to a well-motivated problem.  There was a consensus that the paper is clearly presented (after the rebuttal period; one reviewer initially had some clarity concerns).

(c) Weaknesses: The situation studied is a very special case.  One reviewer also had concerns about the practical efficiency of the algorithm compared to its asymptotic scaling.

**Note From Pc:**

if the above contains the word "oral" or "spotlight" please see: "oral" presentation means -> notable-top-5% and "spotlight" means -> notable-top-25%. As stated in our emails, we are disassociating presentation type from AC recommendations

**Summary Of Ac-Reviewer Meeting:**

n/a